# The Importance of Implementing a Transition Strategy for Patients with Muscular Dystrophy: From Child to Adult—Insights from a Tertiary Centre for Rare Neurological Diseases

**DOI:** 10.3390/children10060959

**Published:** 2023-05-28

**Authors:** Maria Lupu, Mihaela Ioghen, Radu-Ștefan Perjoc, Andra-Maria Scarlat, Oana Aurelia Vladâcenco, Eugenia Roza, Diana Ana-Maria Epure, Raluca Ioana Teleanu, Emilia Maria Severin

**Affiliations:** 1Clinical Neurosciences Department, Peadiatric Neurology, Faculty of Medicine, Carol Davila University of Medicine and Pharmacy, 020021 Bucharest, Romania; maria.lupu@rez.umfcd.ro (M.L.); radu-stefan.perjoc@rez.umfcd.ro (R.-Ș.P.); andra.scarlat@stud.umfcd.ro (A.-M.S.); oana-aurelia.vladacenco@drd.umfcd.ro (O.A.V.); eugenia.roza@umfcd.ro (E.R.); raluca.teleanu@umfcd.ro (R.I.T.); emilia.severin@umfcd.ro (E.M.S.); 2Department of Paediatric Neurology, Dr Victor Gomoiu Children’s Hospital, 022102 Bucharest, Romania; epurediana@gmail.com

**Keywords:** muscular dystrophy, transition, neuromuscular, multidisciplinary, quality of life

## Abstract

Progress in the field of muscular dystrophy (MD) using a multidisciplinary approach based on international standards of care has led to a significant increase in the life expectancy of patients. The challenge of transitioning from pediatric to adult healthcare has been acknowledged for over a decade, yet it continues to be a last-minute concern. Currently, there is no established consensus on how to evaluate the effectiveness of the transition process. Our study aimed to identify how well patients are prepared for the transition and to determine their needs. We conducted a descriptive, cross-sectional study on 15 patients aged 14 to 21 years. The patients completed a sociodemographic and a Transition Readiness Assessment Questionnaire (TRAQ). We also analyzed the comorbidities of these patients. Our study revealed that only 46.7% of the patients had engaged in a conversation with a medical professional, namely, a child neurologist, about transitioning. A total of 60% of the participants expressed having confidence in their self-care ability. However, the median TRAQ score of 3.6 shows that these patients overestimate themselves. We emphasize the necessity for a slow, personalized transition led by a multidisciplinary team to ensure the continuity of state-of-the-art care from pediatric to adult healthcare services and the achievement of the highest possible quality of life for these patients.

## 1. Introduction

Progressive muscular dystrophy (MD) is a multisystemic disease in which a genetic pathogenic variant causes progressive muscle degeneration. The inheritance pattern of MD can be X-linked, autosomal recessive or autosomal dominant. There are multiple types of muscular dystrophies, the most common one being Duchenne muscular dystrophy (DMD), which is also one of the most severe forms and is characterized by the complete or partial lack of dystrophin protein [1,2,3,4].

Historically, before 1970, patients with DMD had a median survival age of 18 years [1,5]. In addition to muscle loss and weakness, patients with MD present with a significant number of comorbidities (respiratory, cardiac, orthopedic, psychiatric, etc.), which makes disease management far more challenging [2,3,4]. With the recent advances in our understanding the pathophysiology of MD, we currently approach it in a multidisciplinary manner following the standards of care and treatment, which are periodically updated and provide clear guidance on how to manage MD, taking into account each possible comorbidity as well as socio-economic aspects. Since the 1990s, the use of steroids has been the basis of therapy, and ventilation has been implemented on a larger scale in medical facilities. Moreover, the life expectancy of patients born after the 1990s has increased significantly, reaching a median age of 28.1 years [1,5,6].

Despite the fact that the transition from pediatric to adult medical care has been an acknowledged issue for over a decade, preparations for the transition are often delayed until the patient reaches adult age or are ignored altogether. This has a negative effect on the patient’s medical care and quality of life [1]. The medical literature lacks sufficient studies and protocols that stress the importance of developing a thorough plan for the transition process from pediatric to adult medical care. Moreover, is there currently no consensus or established protocol on how to effectively carry out the transition process in a manner that ameliorates patient care and quality of life [1,2,7,8,9,10,11,12,13].

In the absence of step-by-step guidance, patients lack access to the resources needed to organize the transition process themselves. Thus, there is an acute need for a strategy regarding the transition of these patients to the medical services that are intended for adults. Were such a strategy to exist, both the patients and the multidisciplinary team assigned to their care would have a clear understanding of how to manage the situation at hand [1,12,13,14,15].

The aim of our study was to evaluate the readiness of patients with muscular dystrophy for the transition to adult healthcare services and to identify their specific needs. We assessed the level of preparation of patients and their families for the transition process, as well as their confidence in their self-care ability. We also analyzed the comorbidities of our patients and highlighted situations that may require specific approaches. By doing so, we hope to provide insight into how to improve the transition process and, ultimately, enhance patient care and quality of life.

## 2. Materials and Methods

We carried out a descriptive, cross-sectional study. The study included patients diagnosed with muscular dystrophy (Duchenne muscular dystrophy, Becker muscular dystrophy and calpainopathy, a type of muscular dystrophy caused by a pathogenic variant in the CAPN3 gene and formerly known as LGMD2A or limb–girdle muscular dystrophy type 2A [16]), over the age of 14 years in the care of the pediatric neurology department of Dr Victor Gomoiu Children’s Clinical Hospital, Bucharest, Romania, a tertiary rare disease and academic center. Patients and their families provided informed consent to participate in this study, and permission to conduct the study was granted by the hospital ethics committee.

Patients included in the study completed a Transition Readiness Assessment Questionnaire (TRAQ) and a sociodemographic questionnaire (date of birth, rural or urban place of origin). The TRAQ was created to quantify the ability of people between the ages of 14 and 21 years to manage their health. It consists of 20 questions divided into 4 categories: medication management, compliance appointments, monitoring of health problems, and discussion with medical staff [17]. The TRAQ also contains 2 questions about the patient’s motivation regarding their own state of health. Each question is rated on a scale of 1 to 5 (1 indicating “No, I do not know how,” 2 indicating “No, but I want to learn,” 3 indicating “No, but I am learning to do this,” 4 indicating “Yes, I have started doing this” and 5 indicating “Yes, I always do this when I need to”), and an average score is calculated for each section. The final questionnaire score is determined by calculating the average of all the section scores. Patients are then divided into four levels of readiness: 1–2, 2–3, 3–4 and 4–5. Those with total average scores of 1–2, 2–3 and 3–4 require additional transition education [17].

The questionnaire was applied in person during the admission of patients to our clinic for a routine health assessment. We also analyzed their medical charts, focusing on aspects such as comorbidities and the wide range of needs that were to be met. All data analyses were conducted using IBM SPSS Statistics, version 26.

## 3. Results

Twenty-two patients with progressive muscular dystrophy were monitored in our clinic. Fifteen patients with muscular dystrophy between 14 and 22 years of age completed the questionnaire and were included in the study. Complete physical and neurological examinations were performed in our clinic by one of the authors of this paper. The mean age at enrolment was 17 years, with the average age of diagnosis being 5.6 years. Twelve patients had Duchenne muscular dystrophy, one had Becker muscular dystrophy, and two had calpainopathy [17].

All our patients had genetically confirmed muscular dystrophy, determined between the ages of 1 and 12, with five of them having undergone muscular biopsy. The diagnostic gap was 3 years (range, 1–12 years of age).

### 3.1. Comorbidities

Of the 15 patients, 9 were ambulatory and 6 were non-ambulatory. Within the non-ambulatory group, the earliest age at which walking ability was lost was 10 years (mean of 13.5 years of age; range, 10–17 years of age). More than half (8/15, 53.3%) had cardiac disease with onset at a median age of 13.5 years (range, 11–15 years of age). All the patients with Duchenne and Becker muscular dystrophy (BMD) were treated with Angiotensin-converting enzyme (ACE) inhibitors prior to any onset of cardiac symptoms, and one patient with calpainopathy was treated with ACE after the onset of cardiac signs. One DMD patient had beta-blockers administered after the occurrence of cardiac disease. Thirteen patients (13/15, 86.7%) had respiratory involvement, with a median age of onset of 13 years (range, 10–17 years of age). One-third (5/15, 33.3%) required non-invasive ventilation, and one patient underwent tracheostomy. All patients in our study had scoliosis, with 8/15 (53.3%) undergoing physiotherapy and 6/15 (40%) having a clubfoot. All DMD and BMD patients received glucocorticoid treatment. In total, 4 (4/15, 26.7%) had Cushing’s syndrome, 7/15 (46.7%) from obesity. Regarding gastrointestinal issues, one patient experienced constipation and also required gastrostomy. All patients had renal function tests within normal limits. Ten (10/15, 66.7%) patients in our study population presented with emotional disorders, with one having anxiety and depression and two patients having expressive language disorders. The main demographic characteristics and comorbidities of the patients with muscular dystrophy are summarized in Table 1.

### 3.2. Transition Readiness

Table 2 provides a summary of the TRAQ questionnaire results for each category and the overall scores. Less than half (7/15, 46.7%) of the patients in our study had spoken to a medical professional about transitioning prior to the completion of the questionnaire. All the patients who had previously discussed the process of transition of care to adult medical specialties did so with a pediatric neurologist. Of those who discussed the transition, most (4/7, 57.7%) said that the discussion was initiated by the doctor.

Most patients expressed that they feel comfortable enough to discuss their symptoms with healthcare professionals, with 10/15 (66.7%) answering, “Yes, I have started doing this” or “Yes, I have always done this.” Regarding their habits of asking for clarification when needed, a total of 11/15 (73.3%) said they do so either all the time or on occasion. The majority, 13/15 (86.6%), declared that they answer medical personnel/professionals when asked about their health. Only 4/15 (26.7%) patients stated that they feel they have the ability or knowledge to complete a questionnaire regarding their past medical history, including any allergies they might have. Most patients (9/15, 60%) kept track of their appointments in a calendar. The majority (12/15, 80%) sometimes or always communicated their needs to medical specialists, and 9/15 (60%) contacted a doctor when they had concerns about their health. A total of 8 (8/15, 53.3%) sometimes or always contributed to medical decisions regarding their health status, and 5/15 (33.3 %) sometimes or always attended a medical appointment or part of an appointment by themselves.

More than half (9/15, 60%) did not make their own doctor’s appointments. Two-thirds (10/15, 66.7%) complied with the instructions received concerning laboratory tests or other check-ups. However, 8/15 (53.3%) did not contact their doctor about unusual changes in their health. More than half (9/15, 60%) said that they make sure they have a means of transportation available when they need to attend their medical appointments. The majority (9/15, 60%) did not fill a prescription when needed, and 11/15 (73.3%) said that they do not know what to do if they have an allergic reaction to medication. The majority (8/15, 53.3%) did not reorder medications before they ran out. More than half (8/15, 53.3%) were able to explain their medication (name and dose) to healthcare providers, and 7/15 (46.4%) spoke with the pharmacist about drug interactions or other concerns related to their medication.

Nine patients (9/15, 60%) said that they have confidence in their self-care ability, but the median TRAQ score was 3.6 (range 1.3–4.8), as shown in Figure 1. The lowest score was found in the “Managing medications“ category, with a median score of 2.6 (range 1–4.8).

## 4. Discussion

To the best of our knowledge, this is the only study that approaches the transition from pediatric to adult healthcare for patients with muscular dystrophy from a patient-centered point of view. In our cohort of patients, the diagnostic gap was 3 years. In the medical literature, the average time gap between the clinical onset of symptoms and diagnosis is 2.2 years [18,19,20]. During this 3-year time span, the disease gradually evolved at a rate similar to that cited in the medical literature. Thus, the progression of muscular dystrophy had affected multiple systems and organs, leading to severe problems such as malnutrition, heart failure or respiratory failure [1].

The average age of the patients who responded to the questionnaire exceeded 17 years, and the majority of these patients stated that the discussion about the transition started after this age or did not occur at all. This delay has a major impact on the patient, and existing studies show that the optimal age for initiating the transition process is 13–14 years [15]. All the patients who had previously discussed the transition had done so with the pediatric neurologist. Among 57.1% of patients, the discussion was initiated by the doctor and not by the patient or his/her family. Thus, the neurologist appears to be one member of the multidisciplinary team who takes the first step toward managing the transition process [21,22].

Most patients (9/15, 60%) had confidence in their self-care ability, but the real score of 3.6 showed us that they overestimated themselves in this respect. This significant discrepancy may be due to the fact that patients do not understand exactly what the transition process entails. Thus, providing these patients with clear, step-by-step explanations is crucial if they are to be truly prepared for both making the transition and achieving the highest degree of independence possible in their adult lives.

The lowest TRAQ score (median of 2.6) we registered concerned the ability to manage medication and other involved therapies. Therefore, we highlight the importance of thoroughly explaining the characteristics of each supplement, medication and therapy, not only to the patients’ families but also to the patients, and to ensure an in-depth understanding of what these treatments entail.

The best results concerned the ability to talk to the medical staff, with a median score of 4 (range, 1.2–5). This shows us that a close patient–physician relationship based on mutual trust and respect, built over time, provides the patient with a comfortable setting in which they can openly talk with the medical team. With regard to the transition, we emphasize that a slow transfer carried out over a longer period of time is beneficial for building a similar relationship with members of the team dedicated to adults.

Most of our patients considered themselves able to make appointments with the doctor, follow the instructions received from healthcare specialists and contact them if necessary. We believe it is essential to encourage the family to give the patient the opportunity to progress through these steps alone so that the patient will feel and eventually become independent.

### 4.1. Cardiac Management

Cardiac disease is one of the most significant causes of mortality and morbidity in patients with muscular dystrophy [1,23,24,25,26,27]. The cardiac management of the patients in our study was in agreement with the guidelines for managing muscular dystrophy [13,23]. Eight (8/15, 53.3%) of our patients had a diagnosed cardiac pathology at the time of the study. As approximately 30–40% of deaths occur as a result of cardiac disease [1,26], the effective management of heart disease is essential to the care of a patient with muscular dystrophy during childhood and adulthood [13,26,27].

### 4.2. Respiratory Management

Throughout the evolution of muscular dystrophy, a multitude of respiratory comorbidities develop. Therefore, the patients’ monitoring begins early in the disease trajectory and continues throughout their life [13,24,28,29]. During the transition period, when patients usually become non-ambulatory, the need for respiratory interventions appears most frequently and requires non-invasive or tracheostomy-assisted ventilation, assisted coughing and oxygen therapy [13,29]. Among the patients in our cohort, 13/15 (86.7%) had restrictive respiratory failure, 5/15 (33.3%) required non-invasive ventilation, and 1 underwent tracheostomy.

Thus, the transition takes place during a difficult period in which the need for monitoring and respiratory interventions is vital and has a major impact on quality of life and life expectancy [13,30].

### 4.3. Rehabilitation and Orthopedic Management

Eight (8/15 (53.3%) patients followed a regular rehabilitation program with the help of a physiotherapist. For socio-economic reasons and motives related to the compliance of the patient’s family, 7/15 (46.7%) patients underwent medical rehabilitation only during hospitalization. The current guidelines recommend submaximal but constant physical activity for patients with muscular dystrophy [31,32].

All patients (15/15) had scoliosis, 6/15 (40%) had a clubfoot, and 1 required surgical intervention for Achilles tendon lengthening.

Among the consequences of the lack of physical activity are the additional loss of muscle strength, loss of joint range of motion, joint contractures and bone changes such as scoliosis, kyphosis or equinus foot. Moreover, long-term treatment with glucocorticoids increases the risk of bone fragility and osteoporosis. Another consequence of this drug therapy is obesity, which, in turn, negatively impacts the osteoarticular system [33,34,35].

### 4.4. Endocrinological Management

Endocrinological complications appear as a result of both the disease itself and the treatment with glucocorticoids [36,37,38]. All our patients with DMD and BMD (13/15) received treatment with glucocorticoids according to the international protocol and were monitored by an endocrinologist. A total of 4 (4/15, 26.7%) patients had Cushing’s syndrome, and 7/15 (46.7%) had obesity. In the medical literature, the most frequently mentioned complications are linear growth impairment, osteoporosis, hormonal deficiencies and delayed puberty [1,32,36,37]. These endocrinological changes impact the patients not only physically but also emotionally, in addition to the disability caused by the progression of their disease [37,38].

### 4.5. Gastro-Intestinal and Renal Management

As the disease progresses, its consequences are also manifested in the gastrointestinal system as dysphagia and intestinal motility disorders, such as constipation [39,40]. Thus, careful supervision by both a gastroenterologist and a nutritionist is required to prevent these complications and for the patient to maintain an optimal weight [32,39,40]. In this cohort of patients, one patient had constipation and one required a gastrostomy.

Although renal diseases, such as chronic kidney disease, are cited in the specialized literature, our patients had values within the normal limits for blood pressure, urea, creatinine and urinalysis [41].

### 4.6. Psychological and Psychiatric Management

From a psychological and psychiatric point of view, patients with muscular dystrophy are frequently associated with comorbidities caused by the disease itself, owing to the absence of cerebral dystrophin, as well as the devastating impacts of the disease [42,43]. Consequently, these patients are affected by anxiety and depression, and in certain cases, they have learning disorders, autism spectrum disorder and attention-deficit/hyperactivity disorder [44,45,46,47,48].

Among our patients, 10/15 (66.7%) had a diagnosis of an emotional disorder, 1/15 (6.67%) had anxiety and depression disorder, and 2/15 (13.3%) had speech disorders. No patients had either a diagnosis of ADHD or autism spectrum disorder. All patients were periodically evaluated by a pediatric psychiatrist and a clinical psychologist.

The transition process is particularly delicate from a psychological point of view. We believe that it would be ideal for the patients to continue being evaluated by the same psychotherapist with whom they have created a relationship in order to offer them a locus of stability during this major change that they are progressing through.

### 4.7. Socio-Economic Aspects

Whether in written or electronic form, medical records are essential to patient care and safety. Patients and their families are educated on the importance of keeping the medical records as their condition progresses, and they serve as a crucial component in the process of transitioning from a multidisciplinary pediatric team to an adult-focused one. We draw attention to the fact that the transfer of electronic patient records from one institution to another is currently not possible in Romania.

Although aspects of the medical management of the health status of patients with muscular dystrophy are particularly important, their adult lives entail many more challenges. These young people need their status as adults to be accepted, in the sense that they should be able to make decisions regarding the management of both their state of health and aspects related to care, housing, schooling, etc. [1,17,49]. They require 24/7 home care, which is usually provided by family members. However, if provided with adequate financial support and access to care by people outside the family, these patients can benefit from a degree of independence from their relatives, who will also be partially freed from the burden of providing round-the-clock care [1,17].

These young adults must be helped to ensure that they have adequate housing conditions and means of transport and have their opinion taken into consideration regarding the question of whether or not to continue living with the family. In addition, at this point, patients should be made aware of their options for education, which is usually conducted online, and their career prospects [1,17,47].

The transition process should not change well-implemented aspects of patient care. Patients should be able to carry out their lives as they have done before, without needing to make major changes that would pose additional burdens to such a debilitating disease. The transition should start around the age of 12–14 years and should be finalized by the time the patient becomes an adult, which is a challenging period for any child, as the desire for autonomy increases [15]. Among patients with muscular dystrophy, this period is especially difficult, as the loss of ambulation and most comorbidities appear simultaneously with a higher need and desire for independence [50].

The process of the transition from pediatric medical care to adult-focused care must be adapted to adolescents and young adults, taking into account both their needs and their psychological profiles. Despite being young, these patients’ desires should be the central focus, and they must be actively involved in the discussion and decision-making process [12,21,22,44,45,50,51,52].

In Romania, doctors with pediatric specialties can take care of patients up to 18 years of age, after which their care is taken over by doctors dedicated to caring for adults. Thus, the process must be initiated in advance by raising the subject when patients are around 12 years of age and commenced at age 13–14 years [21].

We highlight the need for a healthcare professional with sufficient training in identifying and caring for the needs of patients with MD, as presented in Figure 2. This team coordinator supervises and facilitates the entire transition process while also ensuring that the patients benefit from the multidisciplinary care needed in order to meet their complex requirements [21,22]. In Romania, no such coordinator currently exists; hence, this task relies on the pediatric neurologists. We believe that a training program designed for social workers or family medical doctors that targets this role would ensure overall higher-quality management of the whole process. That being said, the first step in initiating the transition should be the establishment of a coordinator with the agreement of the family. Then, the coordinator should schedule visits to the pediatric wards by the adult medical doctors to meet the young patients and the multidisciplinary team who cares for them while the patients are still under the care of pediatric doctors [21,22].

The transition to a multidisciplinary clinic must be undertaken in such a way that the patient can benefit from all the evaluations needed in a single admission, without having to make multiple trips. Furthermore, to render the whole process easier for the patient, we believe that periodic neuropsychological evaluations and psychotherapy sessions would be extremely useful and should be provided as part of their medical journey [21,45].

### 4.8. Palliative Care

Any chronic disease, especially one that is life-limiting, reaches the point when a palliative care evaluation is a necessity. The palliative care specialist, at the appropriate time, as determined by the patient, the family and the multidisciplinary team, should provide support regarding the management of symptoms and access to hospice services, taking into account the patient’s wishes and values regarding the approach to the end of his/her life [21].

In the context of the increased life expectancy of patients with MD, this topic is an important subject that must be addressed in various moments throughout their lifespans. Advanced care directives and end-of-life discussions still represent a relatively new and under-researched area for individuals with MD. [21].

Romanian pediatric palliative care is a challenging field due to the shortage of palliative medicine specialists, especially for non-oncological patients.

Since we manage our patients in a pediatric setting, the need for a palliation consultation only appeared for one of them. This extremely sensitive subject should first be approached with the help of the patient’s family and his/her psychotherapist and should be centered on the needs of the patient with the aim of improving his/her quality of life.

### 4.9. Study Limitations

Due to objective reasons, analytical conduct was not possible. Important limitations include the fact that the perspective on the transition was based on patient-reported information. Parent- and patient-reported outcomes, however, are an important and recognized source of information for primary and secondary outcomes in randomized trials.

Another limitation is the fact that we do not have a dedicated team for providing palliative care services in our hospital.

### 4.10. Future Directions

Continued application of the same questionnaire to our patients could be a helpful means of assessing the effectiveness of the transition process. We believe that future prospective studies could provide valuable information on the active process of transitioning in real time, one major goal being to establish a viable protocol that can be implemented in different countries.

We encourage further research and advocacy for improving access to specialized palliative care services for pediatric patients with chronic diseases.

## 5. Conclusions

In summary, the process of transitioning from pediatric to adult healthcare services for patients with muscular dystrophy is a critical step that requires careful consideration of each patient’s unique needs and circumstances. It involves a gradual transfer of care from a multidisciplinary pediatric team to adult healthcare providers while ensuring emotional and social support for both the patient and their family. Clear guidelines for this process are lacking, and healthcare specialists must strive to render the transitioning patient centered so as to avoid negative impacts on the patient’s quality of life. Continuity of care and careful monitoring are essential to ensuring a successful transition and improving patients’ long-term outcomes.

The topic of transition is of global significance, and this research study underscores the crucial need for patients to experience a genuine transition, rather than a mere transfer. Our institutional objective, as a tertiary care center, is close collaboration with adult neurologist to establish a comprehensive interdisciplinary transition protocol for cultivating empathy, with the patient as a core element.

## Figures and Tables

**Figure 1 children-10-00959-f001:**
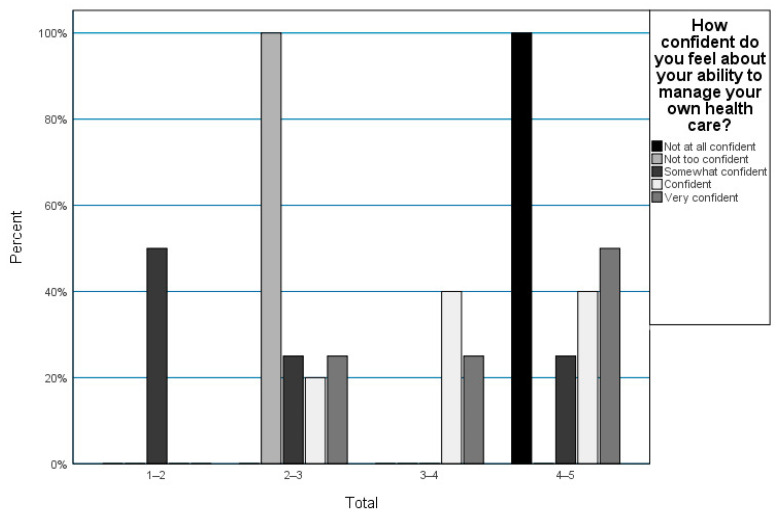
Patients’ confidence in their ability to successfully manage their own healthcare.

**Figure 2 children-10-00959-f002:**
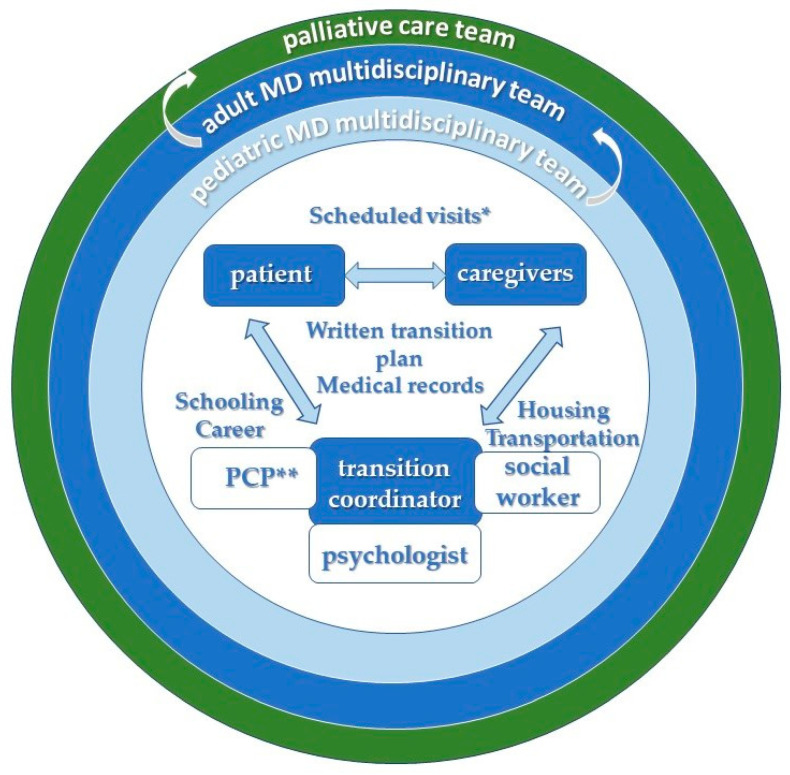
Transition framework from pediatric to adult care for emerging adults with muscular dystrophy. (* Scheduled visits to pediatric wards by adult medical doctors and to adult wards by pediatric patients; ** PCP = primary care provider. The multidisciplinary team is comprised of a neurologist, cardiologist, pneumologist, rehabilitation specialist, orthopedist, psychiatrist, gastroenterologist, nutrition specialist, endocrinologist and nephrologist).

**Table 1 children-10-00959-t001:** Demographic characteristics and comorbidities of patients with muscular dystrophy in the care of the pediatric neurology department of “Dr. Victor Gomoiu” Children’s Clinical Hospital, Bucharest, Romania.

	Summary Statistics
Age (median, range)	17 (14–22)
Age at diagnosis (median, range)	6 (1–8)
Living location (n, %)	
Urban	6 (40)
Rural	9 (60)
Comorbidities (n, %)	
Cardiac	8 (53.3)
Pulmonary	13 (86.7)
Orthopedic	15 (100)
Scoliosis	15 (100)
Club foot	6 (40)
Cushing’s syndrome	4 (26.7)
Obesity	7 (46.7)
Emotional disorder	10 (66.7)
Age at onset of comorbidities (median, range)	
Cardiac	13.5 (11–15)
Pulmonary	13 (10–17)
Mobility (n, %)	
Non-ambulatory	6 (40%)
Ambulatory	9 (60%)
Age non-ambulatory (mean, min-max)	13.5 (10–17)

**Table 2 children-10-00959-t002:** TRAQ questionnaire results for each category and overall scores. The score is between 1 and 5 (1 representing “No, I do not know how” and 5 representing “Yes, I always do this when I need to”).

	Mean	Median	Maximum	Minimum	Range
Managing Medications	2.9	2.6	4.8	1.0	3.8
Appointment Keeping	3.4	3.5	5.0	1.0	4.0
Tracking Health Issues	3.2	3.1	5.0	1.1	3.9
Talking with Providers	4.0	4.0	5.0	1.2	3.8
Total TRAQ score	3.3	3.6	4.8	1.3	3.5

## Data Availability

Deidentified participant data supporting the findings of this study are available for an indefinite period of time from the corresponding author. The data can be accessed by professionals for research purposes by contacting the corresponding author directly.

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
