# Peer review of "The Importance of Implementing a Transition Strategy for Patients with Muscular Dystrophy: From Child to Adult—Insights from a Tertiary Centre for Rare Neurological Diseases"

_children, 2023, doi:10.3390/children10060959_

Round 1

Reviewer 1 Report

Q1: Line 29-30: Duchenne muscular dystrophy (DMD), also being one of the most severe, characterized by the complete lack of dystrophin 

Those sentences are not completely right. Please see your reference 1  “ The etiology of this congenital X-linked disease affecting 1 in 3500 persons, mostly male, are point mutations, deletion, or duplication of gene-coding dystrophin.

Q2: line31-32: Historically, patients suffering from DMD were not expected to survive past 18 years of age in the absence of medical intervention [1].

The original sentences from your reference 1 is “Historically, insufficiency in the cardiorespiratory system caused death before 18 years of age”.

It sounds to me that the original sentences are totally different from yours. As your point in those sentences is “medical intervention, but the reference 1 is the heart disease. If you cite the reference, you should paraphrase, not write what you want to emphasize but deviate the original meanings.

Q3: This makes disease management all the more challenging.

This sentence is full of English errors. For example, it should not be “This”, “which” may be better. It should not be “makes”, because “which” represent a significant number of comorbidities.  “all the more challenging”?? ----“much more challenging”

Q4: line 35:  recent years' advances towards understanding this disease---- recent advances in understanding the pathophysiology(?) of DMD

Q5 line 36:it? What did you mean?  

        the international consensus?  You should briefly introduce “the international consensus” here.

Q6: Considering the recent years' advances towards understanding this disease, nowadays we approach it in a multidisciplinary manner following the international consensus on standards of 36 care and treatment. As a result, the patients' life expectancy has increased significantly, reaching a median age of 28.1 years [1,5,6]

My understanding regarding those sentences is the DMD’ life extend to 28.1 is due to your approaching it in a multidisciplinary manner. This is 100% wrong. Please do a complete revision. I do not think that the authors of the Neurology will be happy if they read these comments.

Q7 40-41: the preparation for said transition is often addressed at the last minute, when the patient reaches the age of 18 years old, if considered at all.

I am totally confused by those sentences and do not understand your points.

Q8 line 41-42 : This has an intensely negative effect on the patient's medical care and quality of life.

Can you tell me and your potential readers where this comment from? from your own observation ( apparently not, because your data showed that patients with DMD had TRAQ score of 3.6). I presume this comment may be from somewhere you have read. Please add them to the end of this sentence.

Q9 line 47-48: patients find themselves almost alone in the face of major difficulties, with no knowledge of how to overcome these challenges themselves

Too many redundant words.  I think that “patients always solve problems on their own” can replace your sentences here.

Q10:line 49-58:need an English native speaker do a complete revision.

Q11: A table regarding the basic data of patients you enrolled in this study is needed, such as the age, sex, diagnosis, TRAQ scores, patient’s condition (ambulatory or non-ambulatory), Comorbidities, Transition readiness….etc. In addition, you should address what is “non-ambulatory, such as on wheelchair, walker……… It is very hard for a reviewer or your potential readers to go through your paper without a chart.

Q12: It is very weak for you to present this study with “only a figure 1”. I believe that this dilemma, the pediatricians need to transfer the patient’s data to the adult neurologists or any physicians. Or, patients move to a new place or a country should be educated to carry with a copy of their medical records and hand them to their new doctors. However, the transition sometimes is difficult. Your study only stands on the patient’s view that is not strong enough to highlight the problems, dilemma, and the difficulties of the “transition”.

Q13:discussion

Not new and no conclusive comments. A work-flow chart is needed to integrate all these problems in DMD patients.

English needs a complete revision.

need a complete English revision

Reviewer 2 Report

I thank the opportunity to review the original manuscript entitled "The importance of implementing a transition strategy for patients with muscular dystrophy: from child to adult - insights from a tertiary center for rare neurological diseases". Some points must be evaluated by the authors at this time: 

1. I suggest authors to review the introduction of their manuscript. For example, in the first paragraph, they describe that "Progressive muscular dystrophy (MD) is a multi systemic X-linked genetic disease (...)", which is not correct as muscular dystrophies are associated with X-linked and autosomal recessive and dominant pattern of inheritance. This is a large group, in which dystrophinopathies represent the main X-linked presentation of muscular dystrophy. 

2. Corticosteroids are not recommended in cases of LGMD. In the text, it appears that several patients with different LGMD subtypes used corticosteroids as well as angiotensin-converting enzyme inhibitors. 

3. One important point regarding the main issue discussed in the manuscript would be to add a table summarizing the main aspects of diagnosis, multidisciplinary care and therapies involved in transition strategies from childhood to adulthood. 

Round 2

Reviewer 1 Report

The authors have addressed the reviewer's problems. I recommend it to be accepted. 

Author Response

We appreciate all your valuable comments and suggestions, which helped us in improving the quality of the manuscript. Thank you!

Reviewer 2 Report

The authors markedly improved the quality of their manuscript after the first round of review. I suggest only authors to correct the first paragraph in which they describe that muscular dystrophies result from mutations in the dystrophin gene, so that only dystrophinopathies result from mutations in the DMD gene. 

Author Response

We would like to take this opportunity to thank you for the effort and expertise that you contributed towards reviewing the article!

We agree with your suggestion and changed the phrase as it follows:

Progressive muscular dystrophy (MD) is a multisystemic disease in which a genetic pathogenic variant causes progressive muscle degeneration.